# An Experimental Investigation of an Open-Source and Low-Cost Control System for Renewable-Energy-Powered Reverse Osmosis Desalination

Evangelos Dimitriou *[ID], Dimitrios Loukatos [ID], Eleftherios Tampakakis [ID], Konstantinos G. Arvanitis [ID] and George Papadakis [ID]

Department of Natural Resources Management and Agricultural Engineering, Agricultural University of Athens, 11855 Athens, Greece; dlouka@aua.gr (D.L.); leyteristam@yahoo.com (E.T.); karvan@aua.gr (K.G.A.); gpap@aua.gr (G.P.)
* Correspondence: vdimt@aua.gr

**Abstract:** Considering the degradation of water resources and the increase in human population, desalination seems to be a promising method for meeting the global water demand, from potable water to plant irrigation needs. The contribution of desalination to the agricultural sector, through the supply of water for plants or animals, is critical because this sector represents 70% of the global water demand. Unfortunately, the desalination process is energy-intensive and subjected to several factors that result in undesirable fluctuations on quality/quantity of product water, and/or energy waste. Renewable energy sources can supply the necessary power, but they amplify these challenges because their availability varies over time. A simple and efficient way to tackle this issue is to control the pressure of the feed water before feeding it to the membrane. Typically, the pairing control systems are quite expensive or lack the necessary customization freedom that could improve their operation. Therefore, this study highlights the feasibility of enhancing a typical desalination control equipment via the incorporation of modern low-cost microcontrollers and flexible open-source software; the potential of these tools has not yet been fully explored. The microcontroller executes customized PID logic, driving an industrial inverter module. Our results indicate that the proposed system can keep pace with the desalination process setpoints, reducing the stress of the electromechanical components and periods of out-of-specification freshwater production. This low-level control function implementation minimizes the need for human intervention while providing a promising foundation for further extensions and customizations in this area.

**Keywords:** desalination; automatic control; renewable energy sources; embedded systems; Arduino; sustainability; open-source tools

## 1. Introduction

Projections show that the global human population will reach 9.7 and 11.7 billion in 2050 and 2100, respectively [1]. Therefore, there is great concern about the capacity of the land to meet the rising food demand. According to recent studies, food demand is expected to increase by 98% by 2050 [2]. At the same time, increased food production will require greater inputs of water, energy, or both. Global water demand has increased substantially in the last 20 years, and it is estimated to increase by 55% by 2050. Furthermore, more than 85% of human water consumption is used for agricultural purposes [1,3]. However, climate change and the continuous increase in greenhouse gas emissions will affect water management due to increased variability in natural processes [3]. Hence, it is imperative to find sustainable solutions to ensure an adequate supply of freshwater resources and significantly reduce greenhouse gas emissions. A sustainable solution for freshwater production is a reverse osmosis (RO) desalination system powered by renewable energy sources (RES) [4,5].

RO desalination is an energy-intensive process due to the high pressure required to overcome the osmotic pressure for producing potable water [6]. Conventional RO desalination units are combined with energy recovery devices, drastically reducing the specific energy consumption of the RO units. Furthermore, in order to reduce the use of fossil fuels and their impact on the environment, RO desalination units can be integrated with RES technologies to find an environmentally friendly solution for the production of freshwater [7]. In the literature, several RO desalination plants powered by RES, such as photovoltaics (PV), wind turbines, or sea waves, have been studied and proposed [4,5,8]. However, the combination of RO desalination units with PV, wind turbines, or sea waves, is problematic due to the inherent uncertainty and intermittent nature of RES and because it requires large and costly energy storage devices to maintain a continuous and constant power supply to RO units. Hence, electric storage should be minimized or eliminated, if possible, while operating the RO desalination units under variable operating conditions (transient flow rate and pressure) attributed to the instantaneous renewable power available [9–11].

Experimental studies showed that a direct connection of RO units with RE technologies could result in lowering specific energy consumption due to the partial load operation of the desalination unit [7–9]. Variable operation of the RO unit can be succeeded with a Variable Frequency Drive (VFD), which controls the speed of the High-Pressure Pump (HPP), and it is utilized according to the needs of the power system [12]. Hence, an energy management system is necessary to continuously regulate the operation of the desalination unit by taking into account several variables and parameters, such as the instantaneous RE power available, the water demand (freshwater flow rate), as well as the operational limits (membrane inlet pressure and quality of product water) of the desalination unit [13]. There are several reports in the literature where desalination systems are equipped with control systems, showing excellent results in the overall management of the RO unit [9,13]. However, the energy management and control systems that have been used in the desalination unit are expensive and inconvenient to use and require expert staff for their installation and configuration. Indeed, the majority of these systems utilize comparatively expensive industrial equipment and programmable logic controller units (PLCs) that lack the necessary programming flexibility. In addition, their fluent cooperation with additional motor driving modules (e.g., inverters) is elementary, or their cost exceeds the limited budget allocated for a small desalination unit.

To find suitable alternatives, the proposed work investigates the feasibility of utilizing innovative, open-source, and low-cost hardware and software tools for supporting the critical low-level control functions of a typical desalination unit powered by renewable energy sources. This alternative is a cost-effective and user-friendly low-level control mechanism that allows an RO desalination unit, powered by photovoltaic (PV) systems and/or wind turbines, to operate under partial load conditions, even when the power supply from RES is not optimal. For instance, variations in wind speed or partial solar obstructions, such as cloud cover, can impact energy availability. Despite these fluctuations, the system ensures the production of fresh water at an acceptable quality level. Additionally, the control system effectively stabilizes fluctuations in membrane inlet pressure, which may arise from changes in feed water temperatures, in addition to the inherent variability of renewable energy sources. This stabilization preserves optimal membrane performance and, consequently, overall system efficiency. The tools utilized for this purpose mainly include well-documented and software-supported open-architecture microcontrollers, small actuators, and standard sensors. This equipment, combined with the preexisting one, can offer the necessary freedom to collect the critical physical parameters and fine-tune the operation of the desalination unit on a continuous basis, thus following the product water requirements. This work also describes how the basic low-level control mechanism can be connected to more advanced complementary computation and communication modules, either locally or remotely.

However, it is worth noting that the applicability of similar arrangements is not apparent. Indeed, despite the diverse benefits of modernizing industrial processes [14],

many people in this sector face barriers to making progress in this area, which is partly due to the low level of education, unawareness of the potential for improvement through Industry 4.0 adoption, and fears of high implementation costs [15]. As outlined in this article, the need for retrofitting, i.e., adding new technology or features to older systems, has been recognized by many leading automation companies, while, at the other end of the spectrum, small customized microcontroller-based boards, although in its infancy stage, are cost-effective and contribute to the demystification and adoption of cutting-edge solutions toward Industry 4.0. In this regard, one of the main priorities of our work is to highlight the feasibility of embedding widely available microcontrollers to retrofit important real-world process control operations, such as desalination, at low cost and with satisfactory efficiency and flexibility, which could not be provided by the typical black box PLC solutions until now.

The first set of experiments indicated that the applicability of the proposed methods is possible and delivers satisfactory results as the modified system can dynamically follow the setpoints implied by the desalination process, with satisfactory accuracy in time and magnitude. The approach presented in this work uses simple principles and techniques, is customizable, and can be enriched with further inputs and functionality to better support the system hosting it.

The automatic regulation mechanism being presented will allow the desalination system to continue to work on marginal photovoltaic or wind turbine power supply by regulating it to function at lower pressure setpoint levels. Without this functionality, the system would be powered down automatically by the protection circuits of its power supply/conversion equipment. Similarly, by making the desalination system work on partial load and not at its maximum on very sunny or windy days, it will be protected from damage. For intermediate power supply conditions, using the proposed mechanism, the system will be able to accurately follow the product water quality specifications as these are translated into feed water pressure setpoint tracking actions despite temperature fluctuations and component performance degradation over time.

The rest of this paper is comprised of the following sections: Section 2 provides a more detailed aspect of the challenges and motives behind this work and highlights the corresponding design and material specifications. Section 3 explains the main parts of the system being upgraded. Section 4 highlights interesting implementation step details regarding the proposed system. Section 5 describes the necessary evaluation setup and discusses the results. Finally, Section 6 contains concluding remarks and directions for future investigation.

## 2. Motives and Challenges

The system proposed in this paper can contribute to finding solutions to some of the most pressing societal challenges, such as water scarcity as well as sustainable and cheap electricity. Only three percent of the world's water is freshwater, and 66 percent of that is found in frozen glaciers or is unavailable for use [16]. Inadequate sanitation is also a problem for 2.4 billion people; they are exposed to diseases, such as cholera and typhoid fever, and other water-borne illnesses. In addition, the COVID-19 pandemic has highlighted the difficulty of providing billions of people with clean drinking water and sanitation facilities to prevent the spread of the virus.

As a result, water is at the core of sustainable development and is closely linked to poverty reduction and climate change. Great emphasis must be placed on water management and irrigation efficiency and ensure that clean water can be provided to all communities, especially those that are poor, marginalized, and vulnerable. Sustainable Development Goal 6 (SDG 6) on water and sanitation, adopted by the United Nations (UN) Member States at the 2015 UN Summit as part of the 2030 Agenda for Sustainable Development, provides the blueprint for ensuring the availability and sustainable management of water and sanitation for all.

Simultaneously, coal, oil, and natural gas remain the primary global energy sources, even as renewable energy has been increasing rapidly [17]. Over 75 percent of the energy supply worldwide is derived from coal, oil, and natural gas, according to statistics on global energy sources [18]. Meanwhile, the prices of electricity are increasing sharply annually, and recently, there was a 10 percent increase after only one year (2020 to 2021) [19]. In addition to this, the associated taxes and levies have also been increasing.

Hence, it is imperative to find efficient methods for clean and low-cost water production of acceptable quality and quantity, ensuring a sustainable future for everyone. Desalination units running on renewable energy sources seem to offer such a solution, but they also introduce new challenges that need to be addressed.

Typically, a desalination system, capable of operating on renewable energy, is comprised of a water tank, a feed water pump, a pretreatment system, and one membrane module. As water is forced to pass through the membrane, its salinity is reduced, thus making it suitable for the tasks being set. These tasks are translated to quality and quantity requirements that may vary drastically [13] as, for instance, different salinity levels and water amounts are acceptable for plant irrigation and for drinking by animals or humans. The efficient deployment of a desalination mechanism presupposes continuous control (i.e., measurements and adjustments) of its operation because its performance may alter for a variety of reasons, thus resulting in water quality unsuitable for the purposes being set or causing power waste. For instance, if the RO desalination system is directly connected to solar photovoltaics and/or a wind generator, sudden changes in solar irradiation and wind speed result in sudden variations in the available power for the motor of the water pump. Thus, if the pump is underperforming (i.e., underpowered), the water being produced will have higher salinity than desired. Similarly, if the pump is overperforming (i.e., overpowered), the water will be of better quality than needed, and thus the excessive and potentially valuable energy will be wasted. The performance of the membrane may also vary according to the incoming water salinity and/or temperature, while its efficiency becomes lower over time. Finally, the water pump system itself is also subjected to progressive performance degradation due to mechanical and electrical fatigue.

Unfortunately, trying to counterbalance all the abovementioned distortion factors in an automatic manner can be a challenge of high complexity and cost, beyond the potential of a small desalination unit. To reduce the cost, a simple and efficient way to align the quality of the water being produced with the necessary standards is to keep the pressure it has at a specific level before feeding it to the membrane. However, even the implementation of a matching simplistic mechanism has its own difficulties and cost barriers, especially when trying to utilize fully commercial product solutions, which are typically costly and follow the "black box" approach that leaves little freedom for in-situ customizations and adjustments by the ordinary personnel after purchase.

Thankfully, due to the rapid progress in modern electronics, microcontroller modules of satisfactory efficiency are available at a low cost. These modules are accompanied by abundant software offering programming options with wide flexibility. Companies like Arduino or Espressif provide such products for a few dollars. The growing interest in these microcontrollers by engineers and researchers for solving a wide range of simple practical problems related to the control of physical processes favors the expansion of their applicability for tackling more complex ones. The potential that these modern tools have has not yet been fully explored in the area of desalination. More specifically, the engagement of these products exhibits IoT solutions that, in most cases, provide remote inspection of the underlying plant process and/or on-off control of its main inputs/outputs [20–23]. These microcontroller products can be further exploited to implement the main control mechanism for producing freshwater of specific quality via cooperation with pre-existing and more conventional power regulation equipment.

## 3. Materials and Methods

### 3.1. Reverse Osmosis Desalination System

The seawater desalination unit, being available for experimentation and improvements as highlighted by this work, is a small-scale unit with a capacity of 150 L/h and consists of a mixing tank, a feed water pump, a pretreatment system, and one 40–40-inch spiral wound seawater Filmtec membrane module. The unit is also equipped with a hydraulic energy recovery device of the Clark pump type, which is a hydraulic piston pump and replaces the high-pressure pump in a conventional desalination unit. The system works in a closed water loop circuit to avoid continuous solution preparation. A detailed overview of the sub-systems and the components of the Sea Water Reverse Osmosis (SWRO) desalination unit is given in Figure 1.

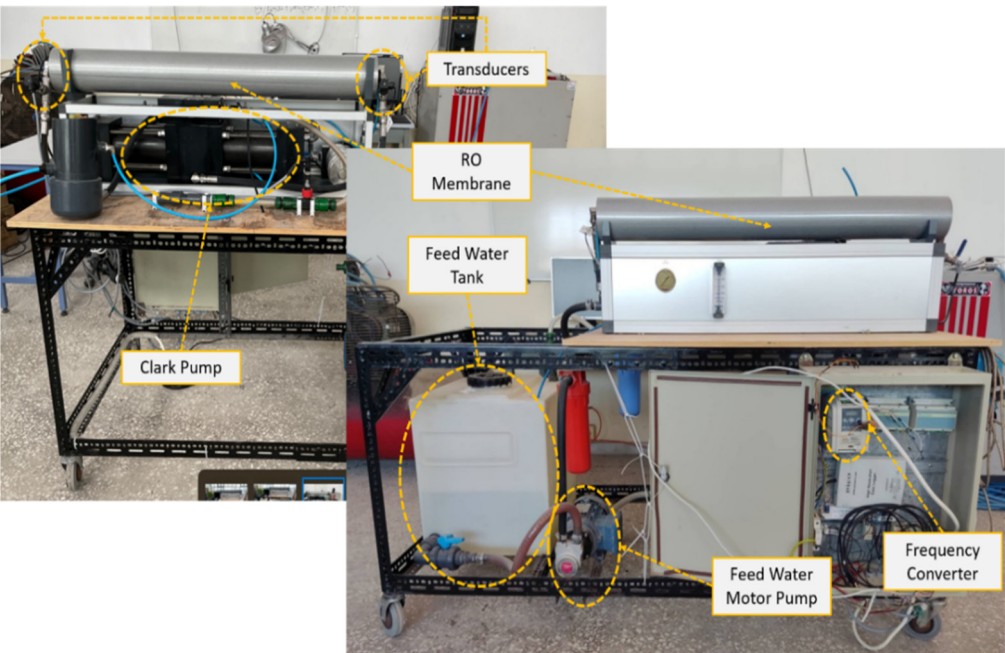

**Figure 1.** Detailed overview of the experimental SWRO desalination unit.

A polyethylene tank with a capacity of 100 L was filled with feed water, an NaCl solution which was prepared by the de-chlorinated tap water. The electrical conductivity of the feed water was adjusted to 50 mS/cm, simulating the seawater. The average feed water temperature was 18 °C.

The desalination unit was equipped with a cellulose 5-micron filter for the feed water filtration in order to increase the efficiency and the lifetime of the RO unit. It is worth mentioning that a cellulose carbon filter was used for the dichlorination of the tap water before it reached the feed water tank the first time that the solution was prepared.

The feed water motor pump assembly transferred the feed water from the mixing tank to the system and provided the positive pressure required at the inlet of the Clark pump. The feed water motor pump assembly consisted of an AC motor and a positive displacement rotary pump. The technical specifications of the motor pump assembly are shown in Table 1.

**Table 1.** Technical specifications of the feed water motor pump assembly.

| Parameter | Value |
|---|---|
| Pump type | Positive displacement rotary |
| Maximum pressure | 17 bar |
| Rated flow rate at 1400 RPM | 1 m3/h |
| Rated power | 0.75 kW |
| Voltage | 1-phase, 230 V, and 50 Hz |

As mentioned before, the hydraulic energy recovery device of the Clark pump replaces the high-pressure pump in a conventional desalination unit. The energy recovery device is a pure mechanical component that has the function of amplifying the pressure supplied by the feed water pump and recouping the hydraulic energy back from the membrane. More specifically, the feed water is pressurized to one of the two cylinders of the Clark pump. The high-pressure brine enters the second cylinder of the Clark pump and exchanges its hydraulic pressure; the result of these actions is the intensification of the feed water pressure to the required membrane pressure (around 50–60 bar). The technical characteristics of the Clark pump are shown in Table 2.

**Table 2.** Technical characteristics of the Clark pump.

| Parameter | Value |
|---|---|
| Type | Clark pump |
| Rated feed flow rate | 1080 L/h |
| Product water flow rate | 150 L/h $\pm$ 15% |
| Rated operating pressure | 55 bar |
| Rated operating feed pressure | 11 bar |

Furthermore, the SWRO desalination unit consists of a spiral wound seawater Filmtec membrane element. The membrane separates the feed water stream into two output streams: permeate and brine. Both streams are driven to the water tank for continuous solution preparation. The RO membrane technical specifications are shown in Table 3.

**Table 3.** RO membrane specifications.

| Parameter | Value |
|---|---|
| Membrane type | Filmtec SW 30-4040 |
| Maximum operating pressure | 69 bar |
| Maximum operating temperature | 45 °C |
| Maximum feed flow rate | 3.6 m$^3$/h |
| Product water flow rate | 300 L/h |
| Salt rejection | 99.4% |
| Single element recovery | 8% |

The desalination unit is also equipped with different sensors and transducers in order to control and manage the operational parameters, ensuring the smooth operation of the system. Hence, the desalination unit is equipped with the following transducers:

- Three analog pressure transmitters (WIKA, A-10) to measure the high-pressure water pressure before and after the membrane element (membrane inlet and outlet pressure) in the range of 0–100 bar, as well as the feed water pressure in the range of 0–60 bar.
- A digital flowmeter (Greisinger, FHKK—PVDF) to measure the permeate flow rate in the range of 0.03–5 L/min.
- An analog flowmeter (Sika, VTH 15) to measure the brine flow rate in the range of 2–40 L/min.
- Two inline conductivity sensors (Greinsinger, GLMU 200, MP) to measure the electrical conductivity of the brine water (0–200 mS/cm) and the product water (0–2000 µS/cm).

Finally, the feed water pump motor of the desalination unit was connected to a power module (frequency converter). This module is an inverter of variable frequency drive (VFD) type unit capable of driving a 3-phase electric motor by modifying the frequency and voltage characteristics of its power supply. This module serves to apply operating point changes for the desalination system by altering the speed of the feed pump and thus affecting the performance of the Clark pump. The inverter is wired between the power source and the feed water motor pump assembly. Modifications in the frequency of the feed water motor pump cause changes in pressure and flow rates at the Clark pump, which in turn affect the inlet membrane pressure. This highlights the interdependence of RO units and energy source systems as variations in the electric power supply can impact the performance of the membrane [7,9,24]. The technical specifications of the frequency converter are shown in Table 4.

**Table 4.** Frequency converter (inverter) specifications.

| Parameter | Value |
| --- | --- |
| Type | Variable Frequency Drive (VFD) |
| Model | HYUNDAI, N50-015SF |
| Applicable motor capacity | 1.5 kW |
| Rated output current | 7 A |
| Rated output voltage | 3-phase, 230 V |

### 3.2. Upgraded Control System Description

The typical small-scale desalination system described in Section 3.1 constitutes the basis for the development and testing of the proposed automatic control mechanism, which is able to follow the membrane inlet pressure setpoints as defined by the desired freshwater salinity levels to produce water of the desired quality. Figure 2 depicts the corresponding hardware upgrade setup.

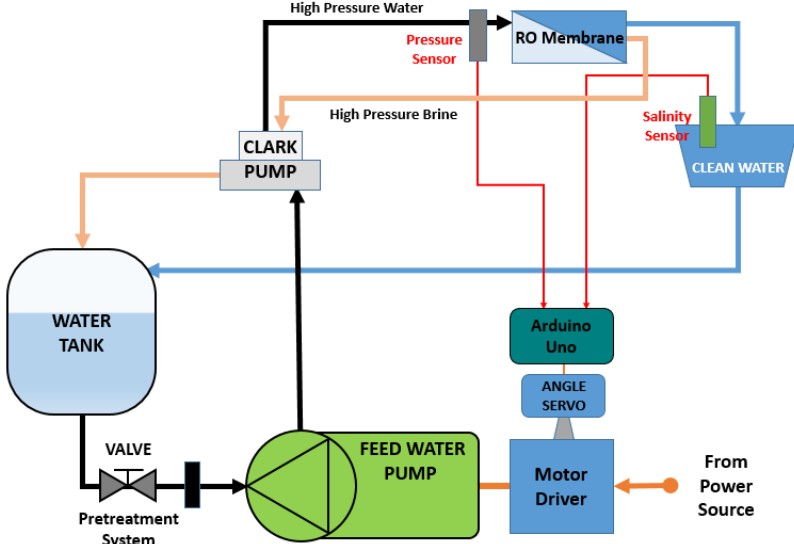

**Figure 2.** The SWRO desalination control system upgrade with new hardware components.

In the first stage, an Arduino Uno unit [25] was utilized as the main microcontroller for performing cycles of typical input reading, processing, corresponding output adjustment, and feedback. This microcontroller is very well supported by exemplification and libraries that facilitate its use, thus being the basis for small project deployment. The accompanying programming environment, called Arduino IDE [26], allows for easy monitoring of the microcontroller in action via the Serial Monitor and the Serial Plotter tools.

The exploitation of the Arduino Uno, by the discussed system, presupposes its involvement in the necessary sensing and acting actions. More specifically, the latter system

has water pressure sensors pre-installed to monitor its operation. These sensors act as transducers that convert water pressure into current, which is converted to voltage drop by connecting an in-series resistor of known value.

The main system contains a motor driving circuit (i.e., the VFD inverter) able to provide manually adjustable power output. The inverter module does not incorporate any desalination-specific intelligence and human intervention but is necessary for keeping the overall process close to the desired levels. Elementary assistance can be provided by a typical PLC unit for turning the system on/off for safety if mechanical stress limits are exceeded. The specific inverter allows for the selection of up to 16 different pump motor speed values via 4 digital control pins and an additional adaptation circuit or for fine-grained frequency changes (i.e., by increments of 0.01 Hz), typically using a potentiometer. The latter method is simpler and more general, allowing for a wider set of performance testing options and was thus adopted by our research approach. As the power that is feeding the water pump can be adjusted via a rotation button (potentiometer), this potentiometer is the key element for applying the automatic control functionality supported by the low-cost microcontroller.

More specifically, the Arduino Uno unit, using the membrane inlet water pressure as input, adjusts the position (i.e., the rotation angle) of an angle servomotor, whose axis is connected with the abovementioned potentiometer. Using these arrangements, the power toward the pump and thus, the water pressure, follow the rotation angle of the button (potentiometer) of the driving circuit. In this way, automatic control functionality is added to a comparatively simple and less expensive desalination system using reverse osmosis membranes.

It must be noted that the fluent operation of the upgraded system requires proper calibration of the pressure sensors and the servomotor-potentiometer mechanism. Furthermore, classic out-of-the-box proportional, integral, and derivative (PID) control techniques [27] are combined with empirical formulas to guarantee satisfactory system behavior. Further details are given in Section 4.

## 4. Implementation Details

### 4.1. Typical Industrial Solution Setup

According to a more conventional hardware setup, utilizing industrial programmable logic controllers (PLCs), the transducers of the basic system function as inputs for the controller cube, while the operation of the pump can be regulated so that the water pressure conforms to the target values and the operation limits of the system are not violated. This mechanism operates well, but it is only an on/off control schema that cannot follow the setpoint values to a satisfactory degree. Indeed, the pump cannot be operated at medium speeds; it can only be completely stopped or operate at full speed.

Consequently, the difference between the setpoint pressure and the achieved one should be long enough to prevent continuous activation/deactivation of the water pump, which drastically shortens its life and results in power waste. The long distance between setpoints and the actual state results in product water of unwanted quality. More meticulous industrial control, providing progressive operation of the pump, would be far more expensive. Quite often, the typical final user has limited skills in and/or access to the programming methods and parameters of the PLC that are typically handled by the expert personnel of the company who did the initial installation under a vendor "lock-in" policy [15].

### 4.2. Sensing Arrangements of the Proposed System

The system proposed in this paper can utilize the preinstalled transducers for pressure (and salinity), as mentioned above. The analog values being measured are converted to digital words, of ten bits each, via the analog-to-digital converter in the Arduino Uno unit. The pressure values being calculated are compared against the readings of the gauge built into the desalination system. The differences are almost negligible, as depicted

in Figure 3a. For the salinity sensors, the values originally gathered were compared against the readings of a precise salinity meter placed at the same measuring point. The corresponding conductivity measurements for the inlet to the membrane water, using the Arduino microcontroller system, were linear but had to be corrected by a factor of 0.85, anticipating the systematic error, to better match the reference values of the feed water salinity measurements. This process is graphically explained in Figure 3b. The salinity measurements of the product water matched the reference instrument well, as depicted in Figure 3c, and thus, no correction was necessary.

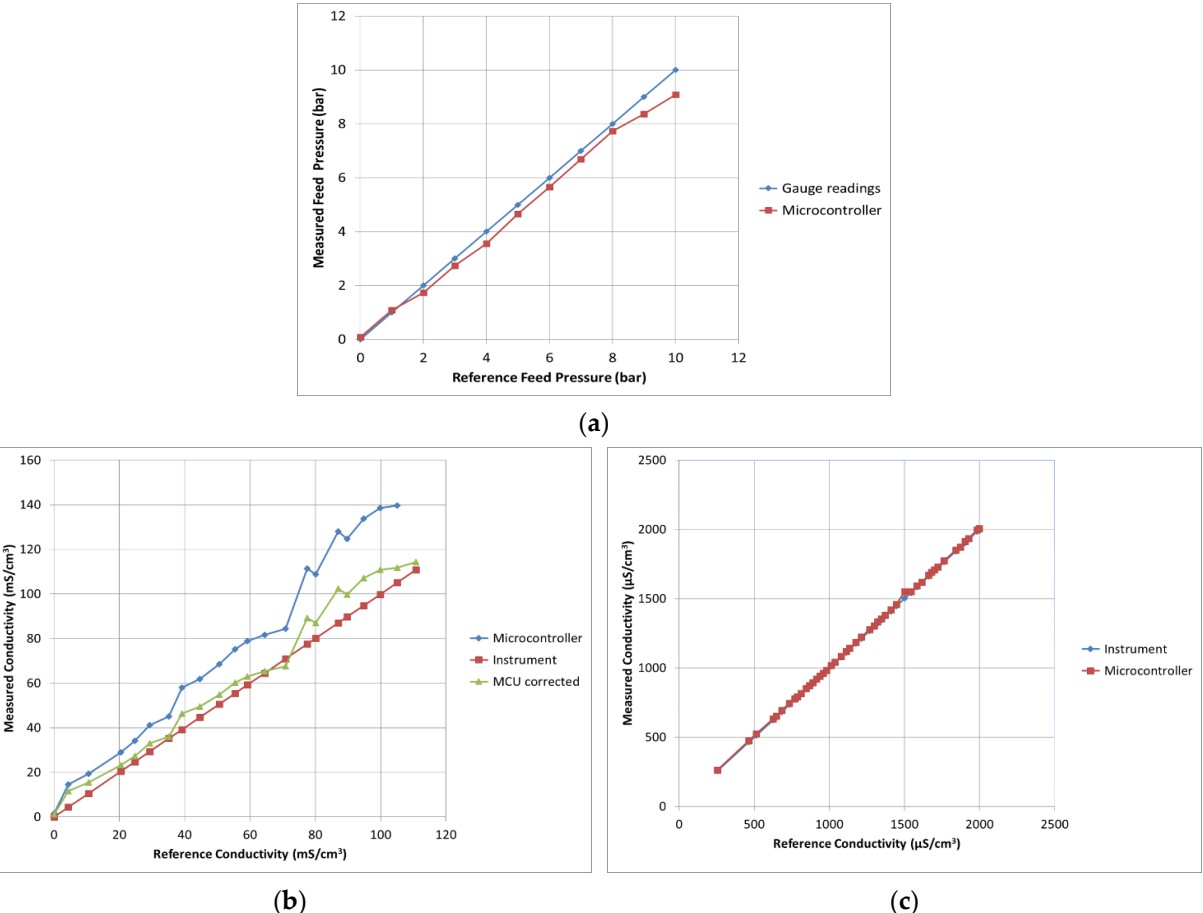

**Figure 3.** (**a**) Inlet pressure measurements, exhibiting a good match with the reference instrument, (**b**) feed water salinity measurements (corrections were necessary to achieve a good match with the reference instrument), and (**c**) freshwater salinity measurements, exhibiting very good match with the reference ones.

### 4.3. Acting Arrangements of the Proposed System

Assisted by the sensing functionality explained in Section 3.2, the Arduino Uno unit senses the ongoing target membrane inlet pressure (setpoint) for the desalination process. These target values were derived through a comparison between the salinity level of the water being produced and the indications of the pressure mechanism. The Arduino Uno unit, comparing the current membrane inlet water pressure (input) with the reference/target one, generates an angle value output for the servo to follow, thus adjusting the operation of the water pump through the rotational button (potentiometer) of the motor driver. The latter button has a rotation range of approximately 270 degrees. However, it is worth mentioning that the pump does not start with non-negligible friction of rotation angle values (nearly the first 25%) of the inverter potentiometer, while the increase in angle in the last 10% also has a negligible contribution to its speed. This is translated to a 65%

area for the 270 degrees (i.e., 175.5 degrees), which can impact the pump's behavior. For this reason, the small servomotor being selected for adjusting the speed of the inverter can be a 180-degree range unit and be placed coaxially with the rotational button of the driving unit, without any loss-of-pump-speed adjustment options.

Figure 4a depicts the exact way that the angle servomotor is coaxially attached to the potentiometer of the motor driver circuit. The servomotor, which is necessary, is small, and thus, its powering needs can be covered by the 5-V supply pin of the microcontroller. The target (pressure) values for the desalination system are provided by the user. A simple method for the latter task, during the initial implementation and experimentation steps, was possible via an auxiliary potentiometer connected to an analog input of the Arduino microcontroller. Figure 4b depicts an instance of the corresponding fine-tuning of the system, utilizing a laptop connected to the Arduino Uno, the Serial Plotter utility of the Arduino IDE software, and a black auxiliary potentiometer for direct target value settings.

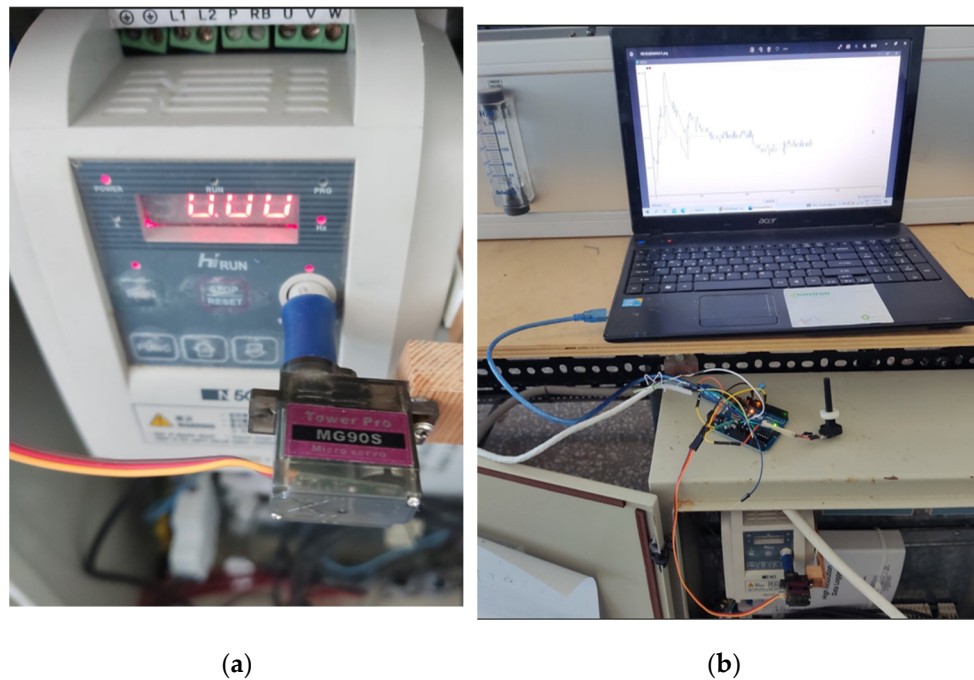

(**a**)  (**b**)

**Figure 4.** Action element arrangements: (**a**) the angle servomotor attachment to the rotational button adjusting the motor driver circuit operation and thus, the pump speed, and (**b**) fine-tuning of the system utilizing a laptop connected to the Arduino Uno, the Arduino IDE software, and an auxiliary potentiometer for direct target value settings.

### 4.4. Algorithmic Behaviour Details

4.4.1. Basic PID Functionality Implementation

The proposed control process exhibits similarities with and thus exploits, the experiences gained through the computerization of an old agricultural ventilation equipment [28]. The basic PID functionality being incorporated into the microcontroller logic is depicted in Figure 5. This control block was implemented in the software, while several tests were performed for tentative use in the RO desalination unit. Assisted by the PID processing block, the reference signal r(t) was corrected according to the error quantity e(t) in order for the modified input to the main system u(t) to produce an output y(t) closer to the desired one. The r(t) is also known as the target or desired process variable or setpoint. The u(t) is the combined output of the proportional (P), integral (I), and derivative (D) parts of the PID module and is also referred to as the control variable (for the main system). The y(t) is also known as the actual or measured process variable.

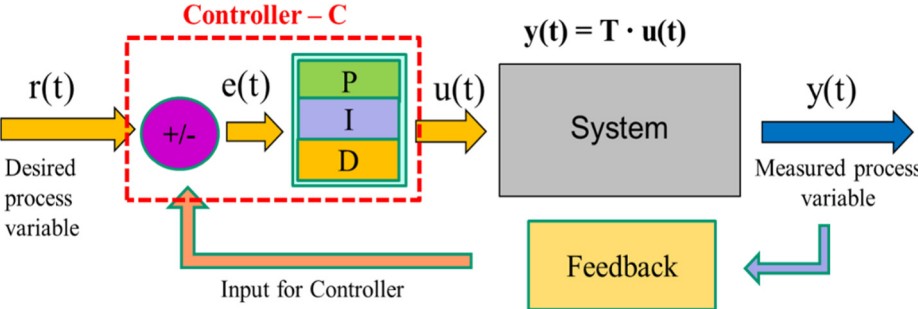

**Figure 5.** Overview of the software-implemented PID controller functionality for tentative use in the RO desalination control system.

The simplified PID mechanism, in the case of the desalination unit, takes into account the target pressure (setpoint) and the current (actual) pressure being achieved for the salt water to pass through the RO membrane (inlet pressure) and adjusts the position (i.e., the rotation angle) of the servomotor, whose axis is connected to the potentiometer in order to minimize differences. The higher the rotation angle, the higher the pressure, and vice versa. The microcontroller provides pulses to finally drive the pump motor properly through the angle servo. These pulses, as number directions to the digital output of the hosting microcontroller, are the output of the PID block and represent the u(t) signal, according to the abovementioned terminology. The salinity values being measured, especially of the product water, are very useful for verifying the performance quality of the overall desalination system and, as shown in Section 4.4.2, for creating precomputed maps of values that can accelerate the whole control process and thus reduce energy consumption and mechanical stress of the system. However, it is not advisable to use them for direct (real-time) guidance of the pump regulation mechanism as their change exhibits longer hysteresis (typically of a few minutes) than the corresponding inlet pressure parameter level alterations. This is an intrinsic characteristic of the RO membrane function, and if followed by the PID mechanism, it would decrease its responsiveness.

In terms of software, the corresponding PID library [29] allows for satisfactory tuning adjustments, such as defining the proportional, integral, and derivative gain behaviors, Kp, Ki, and Kd quantities, respectively, as well as the output limits and the direction (increasing or decreasing) of the corrections.

### 4.4.2. Further Control Logic Enhancements

The system being proposed for desalination utilizes cost-effective and easy-to-program microcontrollers that provide increased flexibility for testing diverse sets of control logic variants. Consequently, the applicability of non-typical PID methods was an interesting option to investigate for achieving cost reduction and faster response of the overall process. As there is no analytical or numerical model describing the relation between the angle values altering the pump speed and the desired pressure, and there are several sources of uncertainty, meticulous experimentation with various automatic control techniques is suggested. These techniques can provide a better performance, like a faster response, shorter oscillations, and less integral windup effects [30] by incorporating non-linear improvements. In this regard, the non-linear technique of using predefined values that were superposed (i.e., combined using weights) on the output of the pure PID controller to enhance the target tracking process was properly adapted due to the good behavior it exhibited. In other words, the PID controller output was partially complemented by predefined angle values generating a specific pressure level to reach the setpoint faster. Since the system can be prone to over- and underperformance, the capability of the PID controller library to produce both negative and positive values as output was also explored. These predefined values had to be calculated empirically.

To this end, a series of measurements were performed, during which the pump was forced to progressively increase the inlet pressure at the membrane by increasing the rotation angle of the potentiometer knob on the VFD converter while the salinity of the product water was recorded. It must be noted that this process was time-consuming as for each measurement, a few minutes of operation at a specific pressure level were necessary. After that, for a given RO membrane installation, consecutive triplets of values of the type ($S_{init}$, $P_{init}$, and $A_{init}$) were stored in the memory of the microcontroller board as a table. Results indicated that product water salinity ($S_{init}$) is directly related to the abovementioned pressure to be applied to the inlet of the membrane ($P_{init}$) and this pressure is achieved by a rotation angle of $A_{init}$ degrees for the potentiometer of the VFD converter being used. Apart from the salinity, a second dimension for finding the optimal pressure to be applied is the feed water temperature, and thus, more than one table containing elements of the ($S_{init}$, $P_{init}$, and $A_{init}$) type should be generated and stored, corresponding to different temperature levels, resulting in a two-dimensional structure. The total number of elements should be kept low in order to be aligned with the memory constraints of the hosting microcontroller. A $10 \times 10$ array should be a good compromise between accurate operation and memory utilization. The exact process of generating the abovementioned elements, expressing the relation between product water salinity, feed water temperature, and inlet membrane pressure is explained in detail in recent experimental studies [24,31,32]. After this time-demanding but important preparatory stage, the main control algorithm, enhanced with the precomputed values technique, is formed as follows:

1. Start of the Algorithm.
2. For a given target product water salinity level and for a given feed water temperature level, seek in the tables for the servomotor angle value that creates an inlet pressure close to the corresponding one ($A_{init}$).
3. Using the current inlet pressure value generated by the desalination system via the water pump as input, compute the new PID (output) value.
4. Compute the angle value that the servo should turn at as the weighted sum of the $A_{init}$ and the current PID output.
5. Apply this composite angle value to the servomotor and wait for the changes to take effect.
6. Repeat steps three through five until the temperature or target salinity level is changed or the user decides to stop the process.
7. If the temperature is changed or the target salinity level is changed, go to step 2.
8. End of the algorithm.

The adoption of this hybrid method can drastically improve the responsiveness of the control system and is consistent with the solutions proposed by similar research tackling slow-varying natural processes in the agricultural sector. The delay between the application of the freshly calculated PID output and the measurement of the new actual pressure is a critical parameter and should be at least of the order of 1–2 s for smoother operation. In addition, by postponing the PID controller correction application by a few seconds each time the setpoint was changed, the system performance was improved, preventing it from overreacting.

The utilization of easy-to-program microcontrollers and the intrinsic flexibility of software-based solutions for testing numerous diverse algorithmic variants for real-world systems are highlighted in this study. In this regard, Figure 6 depicts a testing variant that partially implements the general algorithm described above based on precomputed angle values for a specific requirement of product water salinity level and for a specific feed water temperature. More specifically, Figure 6a focuses on the initialization stage, while Figure 6b focuses on the repetitive steps stage of the algorithm.

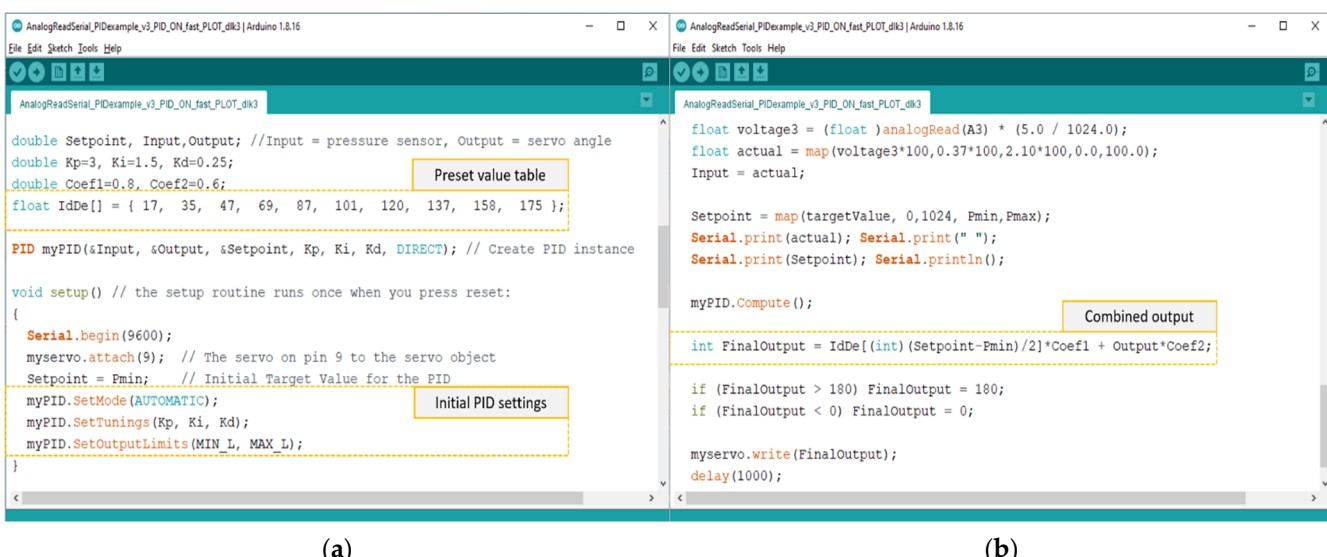

**Figure 6.** Indicative code reflecting a control behavior combining both precomputed and on-the-fly calculated (via PID) output values: (**a**) initialization stage and (**b**) repetitive steps stage.

### 4.5. Facilitating Arrangements

The system proposed in this paper initially utilized an Arduino Uno unit, which is based on the widely used industrial Atmel ATmega8 chip and its successors. The Arduino Uno is well documented, provides stable operation, and has room for experimentation. In addition, its selection for this research is also related to the university's laboratory origins of this work, which favors the use of educationally friendly and well-documented systems. However, the major disadvantage of this microcontroller board is the lack of a remote communication (preferably wireless) interface and its limited memory. These issues can partially be tackled by specifically designed accompanying hardware (i.e., Arduino Wi-Fi shields) or by utilizing a radio module based on the inexpensive ESP8266 chip [33] in a way similar to the one in [28].

For abundant in-situ file storage, visualization, and remote-control functionality, the hiring of an assistive Raspberry Pi single-board computer [34] was the preferred option. Apart from fast remote-control setting changes and efficient storage of the results, easy remote access/delivery of these files [35] is supported, as well as local/remote monitoring and (re-)programming of the low-level microcontroller unit, which remains connected to it via a USB connection as the Arduino IDE software runs fluently of the Raspberry Pi board [28]. The separation between the low-level control functions (performed by a small microcontroller) and the high-level ones (performed by a Raspberry Pi board) to a certain extent is beneficial for security and reliability purposes as well.

To facilitate the experiments, a special application was designed, providing remote monitoring and control functions via communication between the Arduino microcontroller and the mobile phone/tablet device of the user. This application was created utilizing the MIT App Inventor platform [36] and special assistive extensions [37], using UDP packets. A minimal set of messaging between the user's mobile device and the desalination system includes the target value (setpoint) for the inlet RO membrane pressure and the PID controller gains while it provides the actual pressure being achieved as feedback.

## 5. Experimental Results and Discussion

### 5.1. Experiment Setup Details

During the experiments, for a given configuration of the control system variant under testing, the target pressure (setpoint) was altered from comparatively low to comparatively high values, not necessarily in a progressive manner, while the corresponding actual inlet pressure value sequence was recorded at a specific rate. The critical control system

parameters and the setpoints were initially altered via direct code modifications and the assistive potentiometer was installed on the microcontroller, and later installed remotely via the mobile application, which was properly developed utilizing the MIT App Inventor platform. As previously mentioned, the remote-control functionality facilitated the adjustment options, while core Arduino IDE tools, like the Serial Monitor and the Serial Plotter components, provided fast, easy, and detailed inspection of the system dynamics. For these tools to work, the Arduino microcontroller had to be connected to the laptop computer, and later, to the Raspberry Pi unit via a USB connection. The Serial Monitor component can visualize both target and actual inlet water pressure quantities, provided that a pair of such values (separated by space) is written to the serial interface of the Arduino Uno toward the USB port of the computer. The selection of these tools is also justified by the educational origins of this research, which favors the use of simple, user-friendly, and well-documented systems, with reusability potential. The engagement of ESP8266 and Raspberry Pi boards, as explained in Section 4.5, further facilitates this process.

*5.2. System Dynamic Performance*

As mentioned previously, in an RO desalination system that runs on solar or wind energy, drastic changes in weather conditions demand changes to the product water salinity requirements that are often translated to drastic alterations of the inlet pressure target values. The time that the desalination system spends away from its setpoints is translated to water quality that is lower than expected (typically, for pressure values lower than the target ones) or water of better quality than necessary (typically, for pressure values higher than the target ones), resulting in a simultaneous waste of electric energy. Consequently, the off-target time is directly related to water quality deterioration or energy waste. Furthermore, reducing the maximum overshoot and oscillations in system responses is crucial to the desalination unit as it ensures equipment is not unduly stressed and that its lifespan is not shortened. In automatic control terms, elongated periods of off-target operation result in windup problems for the PID controller, while reducing too much the gains signifies a very slow response.

Experimentation with PID values selection indicated that fine-tuning the controller is not achievable by relying solely on classical (e.g., Ziegler-Nichols) methods, for a variety of reasons. First, the common self-oscillation method involves aggressive gain and overshoot and thus, it could cause permanent damage to the parts of the desalination plant, with the RO membrane being the most sensitive. In addition, the step response method, when applied to the system, provided gain values that were rather unsuitable, especially for the proportional term Kp. Indeed, due to the interconnection of the feed pump with the Clark pump unit, the latter inserted strong nonlinearities into the system as it generated sudden fluctuations (e.g., drops during the pressure increase stage) that caused the PID algorithm regulating the power input of the former to overreact.

In this regard, seeking better solutions, a series of experiments were performed to find a good combination of PID parameters offering fast responses, low-amplitude fluctuations, and comparatively short settling times. During these experiments, for the given desalination system, the target pressure (setpoint) level varied between 20 bar and 39 bar while the response of the system (actual inlet pressure) was recorded, typically at intervals of one second. To test a new control logic variant, the Kp, Ki, and Kd gain parameters had to be altered by the user. This empirical approach complemented the difficulties of finding a straightforward regulation solution otherwise. In conclusion, the gain values utilized by the controller had to be considerably smaller than the ones suggested by the step response method (i.e., by at least two or three times). The unwanted windup phenomenon was also reduced that way. In line with the hybrid method of Section 4.4.2, to counterbalance the side effects of the slower system response, empirically precalculated values were superposed on the output of the controller block logic (as expressed in by angle values), accelerating the pressure setpoint tracking process by reducing the corresponding rise and settling times and thus, the intervals of out-of-specification freshwater production.

The utilization of coefficients (weights) applied to both the PID and the predefined/precalculated part of the final output provided further optimization options. It must be noted that the modification of the output limits of the PID, via the method offered by the Arduino PID library [38] was also explored. Indicative results reflecting progressive improvements are shown in Figures 7 and 8. The red line corresponds to the desired target pressure value as implied by the user. The blue line corresponds to the actual inlet pressure generated by the pump, in response to the target directions, as measured by the sensors connected with the Arduino. In all cases, the horizontal axis refers to the time in seconds, while the vertical axis refers to the inlet pressure in bar.

As inferred through inspection and comparison between the graphs in Figures 7 and 8, the response of the control system utilizing customized PID methods must find a good compromise between slow response with long-lasting fluctuations due to integral error accumulations and fast response with agile oscillations that potentially fail to follow the target pressure level. Figure 7 depicts dynamic response traces for the desalination system utilizing coefficient parameters close to the pure PID implementation. Windup and overshoot effects were apparent. As mentioned previously, PID settings had to be quite conservative and thus, a parameter selection close to 3.0, 1.0, and 0.2, for the Kp, Ki, and Kd quantities, respectively, delivered satisfactory results.

The pressure setpoint tracking process was further facilitated by increasing the contribution of the predefined values to the final control algorithm output (i.e., preferably these values were superposed to the PID output after multiplication by a coefficient value close to 0.9). The corresponding performance improvements are depicted by the graphs in Figure 8. The overshoot and the windup effects changed from being significant (i.e., of 20–25%) to almost negligible, and the rise and settling times were drastically shortened; the fall time was also shortened (at least by a factor of three). The presence of the RO membrane introduced strong nonlinearities, indicated by peaks in the graphs, while another interesting observation for all the algorithmic variants being tested was that target pressure values close to the lower limit of the scale (i.e., 20 bar) were more difficult to be reached without small oscillations.

These results demonstrate the feasibility of accurately regulating the behavior of a real system via the embodiment of general-purpose, widely available, well-documented, and low-cost modules based on modern microcontrollers. Indeed, the desalination plant operation point (as it is translated into target pressure setpoint values tracking) can be achieved by executing customized PID logic on the latter microcontroller. The physical quantities acting as feedback for the regulating mechanism can be easily intercepted by the inputs (either analog or digital) of the microcontroller, which, in turn, provides the necessary output to the power converter module driving the feed water pump motor.

The benefits against a blind on/off control operation, typically provided by low-end PLCs, are apparent. Indeed, the desalination system could not operate on partial load conditions matching the (also partial) power availability by RES (e.g., due to variations in wind speed or partial solar obstructions, such as cloud cover) or keep pace with specific product water quality requirements, e.g., due to equipment performance deterioration over time or feed water temperature alteration. Furthermore, on/off actions generate excessive stress on the electromechanical components of the system, reducing their efficiency. The utilization of precalculated values for complementing the output of the pure PID controller block further accelerates the pressure setpoint tracking process, and thus the intervals of out-of-specification water production are further reduced.

Further elaboration is necessary to collect historical data on power consumption and product water characteristics. Similarly, additional salinity, temperature, pressure, and angle measurements are required to fill all the precomputed value tables mentioned in Section 4. The exact profit, in terms of component performance and longevity, product water quality and quantity, and energy economy, constitutes a composite optimization problem that cannot be solved by relying solely on the fidelity of the low-level control mechanism presented in this article. The pattern of the weather conditions alteration, the

water demand profile, or the size of the tanks accumulating freshwater are amongst the factors that affect the overall benefits. There is ongoing research to cover some of these gaps [24].

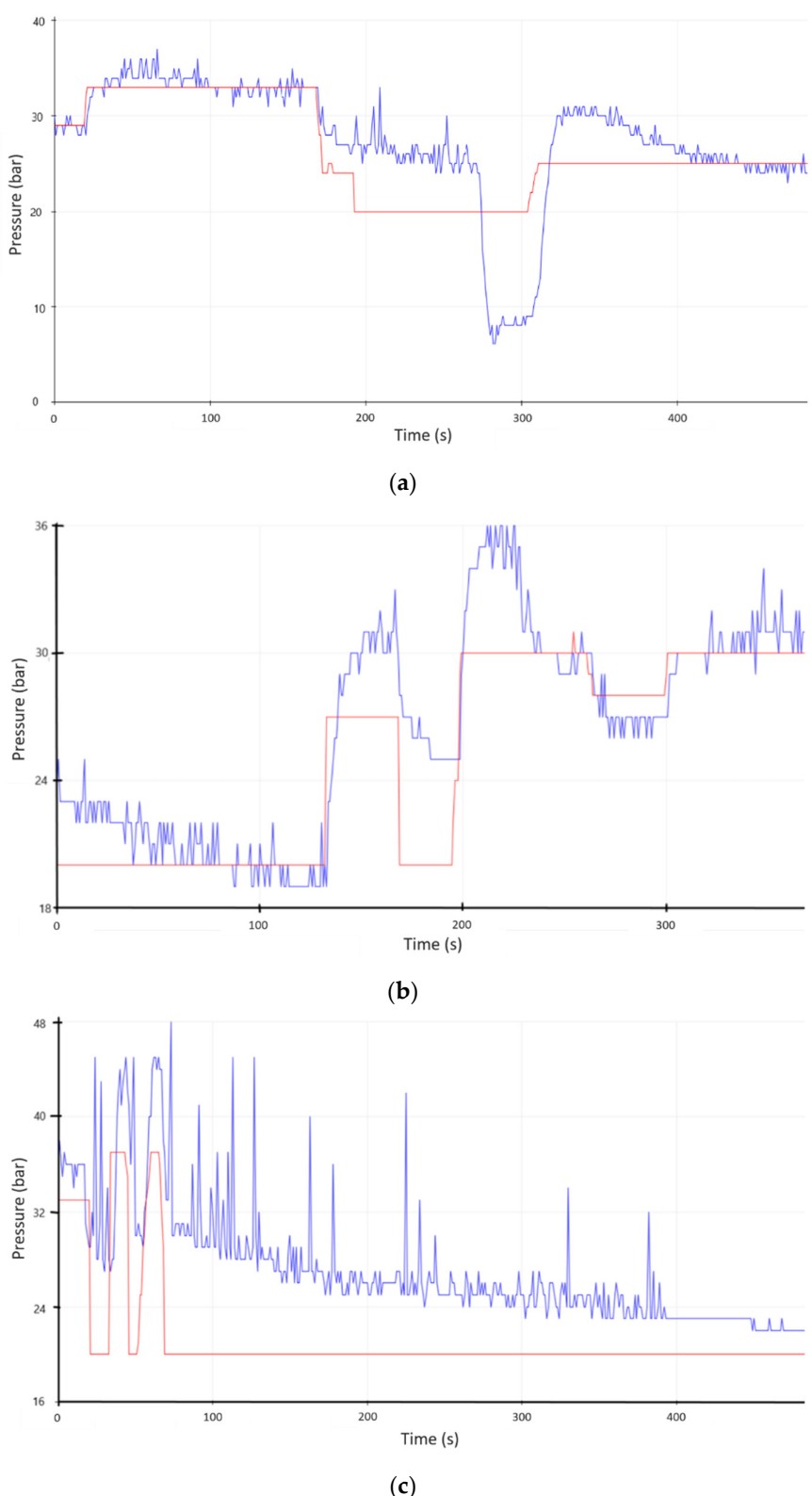

(**a**)

(**b**)

(**c**)

**Figure 7.** Indicative results reflecting the dynamic behavior of the desalination system for different PID settings, revealing performance imperfections mainly due to: (**a**,**b**) overshoot/windup problems or (**c**) slow response (elongated fall time).

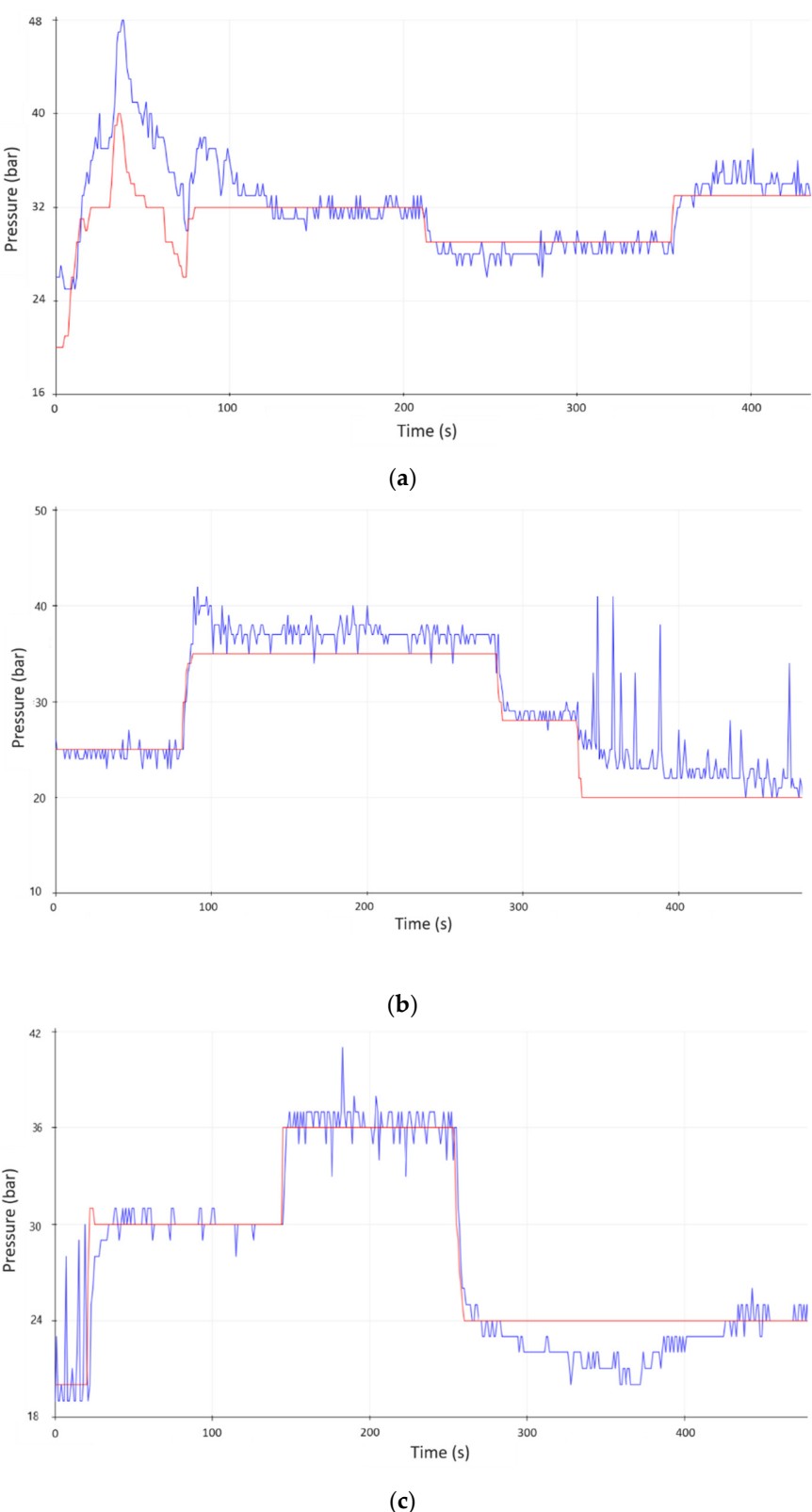

**Figure 8.** Further results reflecting the improved dynamic behavior of the desalination system for different PID settings, assisted by the precomputed pressure values logic for: (**a**) small; (**b**) medium; and (**c**) large alterations of the setpoints.

### 5.3. Financial Cost Issues

The selection of components involved in this research favored cost-effective, easy-to-find, well-documented, and innovative modules. In this regard, for the desalination system, the microcontroller that was utilized in the experiments cost around 10€. The angle servo was 8€, the mechanical arrangements for fixing it to the VFD unit were 3€, and the additional potentiometer and the adaptation resistor for the sensors were 2€. The power supply was 5€, and the remote Wi-Fi radio via the ESP8266 chip-equipped unit was 10€. Thus, the overall cost of upgrading and testing the equipment for the RO desalination unit did not exceed the 40€ limit. The addition of an assistive Raspberry Pi unit and a microSD card would cost an additional 45€ and 5€, respectively.

The market price for a main PLC unit, which is utilized in the industry for the control of a desalination unit, starts at 250€. In addition to this cost, an extra 200 € should be accounted for the extension module required for the communication of two analog transducers with the main unit. Apart from the fact that the cost of a such control system is high, the user should also consider the cost of the desalination system itself, which is also expensive, with a cost of 10,000€, including, pumps, membrane, filters, hydraulic components, and sensors. Therefore, the retrofitting control arrangements utilized in this study have been shown to be a cost-effective alternative solution compared to industrial PLC units, which follow the "black-box" philosophy and at the same time, require expert staff for their installation and configuration.

### 5.4. Further Discussion

The desalination process that utilizes renewable energy sources is a very promising technology but is also energy-intensive and subjected to several factors resulting in undesirable fluctuations on quality/quantity of product water, and/or energy waste. Consequently, the efficient control of the desalination process is of vital importance for its successful adoption. To this end, the system presented in this paper can contribute to achieving fine-tuned control solutions, utilizing widely available and low-cost electronic components and open-source software, providing encouraging guidelines for retrofitting a wide set of real process control cases.

The proposed control logic provides a good balance between performing on-the-fly calculations and utilizing precalculated values stored in the memory. Numerical model techniques could work quite efficiently if based on little but properly chosen data. The low-level control functions presented herein can cooperate with high-level functions (e.g., potentially exploiting weather platform data, product water demand forecast data, optimization, and machine learning techniques) in order to optimize the operation of a renewable-energy-powered desalination system. The Raspberry Pi unit mentioned in Section 4.5 would be sufficient to support these high-level tasks.

The experiences gained with the desalination system modification indicate higher profits due to the minimization of human intervention with the control process at almost negligible additional cost compared to the overall cost of a small desalination plant in rural areas. There are also further cost savings as the electromechanical stress of the participating equipment is reduced and their life is elongated due to the smooth operation policy being followed, and well-maintained RO membranes can result in increased freshwater production efficiency.

On the other hand, several issues remain regarding a more mature implementation version. For instance, further elaboration is necessary with commercial power inverter products allowing for more efficient use of the digital control signals they provide, thus increasing reliability and scalability. Alternatively, the utilization of digital-to-analog conversion circuits should be considered, replacing the angle servo and potentiometer pair for better longevity. In addition, more algorithmic control variants should be tested, and more flexible powering/load equipment must likely be adopted to create and study a wider set of realistic renewable energy supply variation scenarios. The control system of a direct connection RO desalination unit, with PVs and/or wind turbines, installed

in a remote coastal region, could automatically collect data for power production and feed water temperature, thereby optimally adapting its operational parameters, like pump motor speed, membrane inlet pressure, and system feed flow.

It is important to further assess the behavior of the system to various exogenous disturbances that might occur due to the variable nature of the renewable energy sources supplying the small desalination unit. For instance, variations in wind speed or partial solar obstructions, such as cloud cover, can impact energy availability. These disturbances can be studied in the real environment or via emulation scenarios involving special artificial load equipment, partially consuming the power available for the pump. If the changes in power supply due to weather conditions alterations are too big to be counterbalanced by the regulating mechanism, instead of shutting down the system, it is preferable to be able to change the current salinity requirements and to keep pace with the requirements in a rapid and accurate manner, thus continuing to produce water of known quality. To acquire additional benefits, the fast and accurate pressure setpoint tracking function for the RO membrane supporting this functionality can be combined with a logic that is able to redirect product water output to different tanks according to the diverse purposes it is destined for, i.e., potable water for humans and animals or plant irrigation. Thus, another challenging improvement is the engagement of a servo motor (or an electric valve system) redirecting the output of the main desalination system to different tanks according to the acceptable water quality level being produced.

## 6. Conclusions

Desalination system control is typically performed by expensive industrial products that follow the "black-box" philosophy, thus leaving no freedom to optimally adjust their operation to the needs of a specific implementation, thus resulting in energy waste. In response to this situation, this study discusses the enhancements applied to the typical control equipment of a desalination unit, utilizing innovative, low-cost microcontrollers, pre-existing sensors, and cheap actuators to tackle the challenge of keeping pace with the qualitative and quantitative water production requirements as they are expressed by specific membrane inlet pressure setpoints. The proposed improvements utilize open and fully customizable hardware and software components that have already started to be used in analogous situations. Results indicate that the proposed system can dynamically follow the desalination process target values, and it has the potential to support further extensions.

As for future research, the system will be improved to consider more cases of disturbances/imperfections, explore more inputs characterizing the desalination process and thus, form more efficient control methods utilizing a wider range of operating parameters. Such adaptations will ensure that neither the membrane's lifespan nor the specific energy consumption will be adversely affected by the fluctuating power of RES. A commercially oriented version of the proposed techniques, having increased user-friendliness and supporting cooperation with modern network platforms, will also be among the future priorities.

**Author Contributions:** Conceptualization, E.D. and D.L.; methodology, E.D. and D.L.; implementation E.D., D.L. and E.T.; data curation, D.L. and E.T.; writing—original draft preparation, E.D., D.L. and E.T.; validation, E.D. and D.L.; writing—review and editing, E.D., D.L., K.G.A. and G.P.; supervision, K.G.A. and G.P. All authors have read and agreed to the published version of the manuscript.

**Funding:** This research received no external funding.

**Data Availability Statement:** Data are contained within the article.

**Conflicts of Interest:** The authors declare no conflicts of interest.

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
