# Peer review of "An Experimental Investigation of an Open-Source and Low-Cost Control System for Renewable-Energy-Powered Reverse Osmosis Desalination"

_electronics, doi:10.3390/electronics13050813_

Round 1

Reviewer 1 Report

Comments and Suggestions for Authors

The reviewed article largely concerns the important issue of providing cheap and energy-efficient water desalination devices. The authors proposed building the system using cheap and easily available components. I agree with the authors' thesis that it is very important to provide cheap and energy-saving water desalination devices, especially in the context of ongoing climate change. The article lacks information about the type of inverter used to power the pump motor. There are many examples of cheap converters on the market - inverters in which the output frequency can be changed using a digital signal. There is then no need to create an additional drive controlling the position of the inverter potentiometer. However, I would recommend the approach of using the inverter's digital inputs to control its operation. Furthermore, the question arises:

1. Has the shape of the u/f curve been modified in the internal settings of the inverter in order to adapt this characteristic to work with the pump?

2. What is the actual energy gain of the proposed solution?

Author Response

Dear Reviewer #1,

We would like to thank you for your valuable and encouraging suggestions that guided us very efficiently while reassessing the content of our manuscript.

In the attached file, you may find our responses right after your suggestions in different color.

We also attach the revised version of our manuscript, where the same color with each response text is used to reflect the corresponding updates.

Thank for your time - Kind Regards,

The authors 

Reviewer 2 Report

Comments and Suggestions for Authors

Dear authors,

I enjoyed reading this interesting and relevant paper on the experimental work carried out for the control of SWRO systems. The paper is well written in a clear and structured manner, however some significant revisions are required to strengthen the overall quality of the paper:

·       In the introduction there a few relevant and recent references missing related to the paper’s work which I think could very valuable to mention

o   Greco, F., Heijman, S. G., & Jarquin-Laguna, A. (2021). Integration of wind energy and desalination systems: a review study. Processes, 9(12), 2181.

o   Das, T. K., Folley, M., Lamont-Kane, P., & Frost, C. (2024). Performance of a SWRO membrane under variable flow conditions arising from wave powered desalination. Desalination, 571, 117069.

·       Line 209. Regarding the transducer characteristics, please Include specifications of the sensors, range, sample rating, uncertainty, sensitivity/rating?

·       Lines 276-279. The sentence gives a generalized statement, there are any application where this is not the case

·       Line 358-360. Can the authors indicate if this a limitation of the system or of the PID control strategy?

·       Lines 371-373.  Do the authors mean  there is no analytical or numerical model that can relate the mentioned physical variables?

·       Can the authors elaborate how were the parameters of the PID controller selected in terms of performance and or robustness? where there any quantitative indications or comparison with other control techniques?

·       The authors propose an empirical approach to obtain the desired relations between the physical variables. Please elaborate or mention on the possibility or potential to use (existing) numerical models to facilitate this task

·       Figures 7 and 8, the quality of graphs needs to be improved in order to get better readability, please include axis' titles, values and units

·       It is suggested to discuss in more detail the relevance of the system dynamics as it is important to understand the underlying processes an potential effects when upscaling the system to full scale or industrial applications.

·       Section 5.3. How do these costs compare to the cost of the rest of the system?

·       In order to get the paper results into perspective, It would be very valuable if as a reference the authors could elaborate on results with and without the proposed control strategy? perhaps a quantitative comparison in terms of specific energy consumption or fresh water produced?

Comments on the Quality of English Language

no comments

Author Response

Dear Reviewer #2,

We would like to thank you for your warm, accurate and valuable comments that guided us very efficiently while reassessing and enriching the content of our manuscript.

In the attached file, you may find our responses right after your suggestions in different color.

We also attach the revised version of our manuscript, where the same color with each response text is used to reflect the corresponding updates.

Thank for your time - Kind Regards,

The authors 

Reviewer 3 Report

Comments and Suggestions for Authors

The abstract mentions desalination, its importance, and challenges, but it lacks specifics about the study's unique contribution. It should clearly outline the cost benefit from the proposed approach and what type of flexibility it provides. Currently, it is very general. 

The introduction should provide a clear, detailed background on the topic. This includes mentioning of gaps in current knowledge, and a clear statement of how this study aims to fill those gaps.

Fig. 7 is of very poor quality. Please increase font size of the axes. 

In the results section, there's a mention of using an Arduino Uno for data collection. It's important to justify the choice of this instrument over others

The discussion section should address the limitations of the proposed system. This includes potential scalability issues, long-term reliability, and any constraints in the application of the system in different environmental conditions.

While there are mentions of potential for future extensions and customizations, the discussion should be more detailed about specific areas for improvement.

The study lacks novelty but given it's practical nature it could be supported once the above mentioned points are addressed 

Comments on the Quality of English Language

There are issues concerning grammar and overall coherence of the text. Even in the abstract certain parts seem disconnected and the same can be said about the manuscript 

Author Response

Dear Reviewer #3,

We would like to thank you for your valuable remarks that made us to reassess the manuscript, in order to better communicate the contribution of our work. The updated version being submitted contains several new and reformed material.

In the attached file, you may find our responses right after your suggestions in different color.

We also attach the revised version of our manuscript, where the same color with each response text is used to reflect the corresponding updates.

Thank for your time - Kind Regards,

The authors 

Round 2

Reviewer 2 Report

Comments and Suggestions for Authors

Adjustment and modifications to the manuscript have raised the quality and all my comments have been addresed succesfully. Thus, I can recommend the editor to accept the manuscript for publication.

Reviewer 3 Report

Comments and Suggestions for Authors

The authors have revised the manuscript according to my comments 

Comments on the Quality of English Language

It has to be proof read for grammatical errors and coherence